# GRAPH CONVOLUTIONAL REINFORCEMENT LEARNING

**Jiechuan Jiang**[1], **Chen Dun**[2*]**, Tiejun Huang**[1] **& Zongqing Lu**[1†]
[1]Peking University, [2]Rice University

## ABSTRACT

Learning to cooperate is crucially important in multi-agent environments. The key is to understand the mutual interplay between agents. However, multi-agent environments are highly dynamic, where agents keep moving and their neighbors change quickly. This makes it hard to learn abstract representations of mutual interplay between agents. To tackle these difficulties, we propose graph convolutional reinforcement learning, where graph convolution adapts to the dynamics of the underlying graph of the multi-agent environment, and relation kernels capture the interplay between agents by their relation representations. Latent features produced by convolutional layers from gradually increased receptive fields are exploited to learn cooperation, and cooperation is further improved by temporal relation regularization for consistency. Empirically, we show that our method substantially outperforms existing methods in a variety of cooperative scenarios.

## 1 INTRODUCTION

Cooperation is a widespread phenomenon in nature from viruses, bacteria, and social amoebae to insect societies, social animals, and humans (Melis & Semmann, 2010). Human exceeds all other species in terms of range and scale of cooperation. The development of human cooperation is facilitated by the underlying graph of human societies (Ohtsuki et al., 2006; Apicella et al., 2012), where the mutual interplay between humans is abstracted by their relations.

It is crucially important to enable agents to learn to cooperate in multi-agent environments for many applications, *e.g.*, autonomous driving (Shalev-Shwartz et al., 2016), traffic light control (Wiering, 2000), smart grid control (Yang et al., 2018a), and multi-robot control (Matignon et al., 2012). Multi-agent reinforcement learning (MARL) facilitated by communication (Sukhbaatar et al., 2016; Peng et al., 2017; Jiang & Lu, 2018), mean field theory (Yang et al., 2018b), and causal influence (Jaques et al., 2019) have been exploited for multi-agent cooperation. However, communication among all agents (Sukhbaatar et al., 2016; Peng et al., 2017) makes it hard to extract valuable information for cooperation, while communication with only nearby agents (Jiang & Lu, 2018) may restrain the range of cooperation. MeanField (Yang et al., 2018b) captures the interplay of agents by mean action, but the mean action eliminates the difference among agents and thus incurs the loss of important information that could help cooperation. Causal influence (Jaques et al., 2019) is a measure of action influence, which is the policy change of an agent in the presence of an action of another agent. However, causal influence is not directly related to the reward of the environment and thus may not encourage cooperation. Unlike existing work, we consider the underlying graph of agents, which could potentially help understand agents' mutual interplay and promote their cooperation as it does in human cooperation (Ohtsuki et al., 2006; Apicella et al., 2012).

In this paper, we propose graph convolutional reinforcement learning, where the multi-agent environment is modeled as a graph. Each agent is a node, the encoding of local observation of agent is the feature of node, and there is an edge between a node and its each neighbor. We apply convolution to the graph of agents. By employing multi-head attention (Vaswani et al., 2017) as the convolution kernel, graph convolution is able to extract the relation representation between nodes and convolve the features from neighboring nodes just like a neuron in a convolutional neural network (CNN). Latent features extracted from gradually increased receptive fields are exploited to learn cooperative policies. Moreover, the relation representation is temporally regularized to help the agent develop consistent cooperative policy.

---

[*]Work done at Peking University.
[†]Correspondence to Zongqing Lu <zongqing.lu@pku.edu.cn>.

Graph convolutional reinforcement learning, namely DGN, is instantiated based on deep $Q$ network and trained end-to-end. DGN shares weights among all agents, making it easy to scale. DGN abstracts the mutual interplay between agents by relation kernels, extracts latent features by convolution, and induces consistent cooperation by temporal relation regularization. We empirically show the learning effectiveness of DGN in jungle and battle games and routing in packet switching networks. We demonstrate that DGN agents are able to develop cooperative and sophisticated strategies and DGN outperforms existing methods in a large margin.

By ablation studies, we confirm the following. Graph convolution greatly enhances the cooperation of agents. Unlike other parameter-sharing methods, graph convolution allows the policy to be optimized by jointly considering the agents in the receptive field of an agent, promoting the mutual help. Relation kernels that are independent from the input order of features can effectively capture the interplay between agents and abstract relation representation to further improve cooperation. Temporal regularization, which minimizes the KL divergence of relation representations in successive timesteps, boosts the cooperation, helping the agent to form a long-term and consistent policy in the highly dynamic environment with many moving agents.

## 2 RELATED WORK

**MARL.** MADDPG (Lowe et al., 2017) and COMA (Foerster et al., 2018) are actor-critic models for the settings of local reward and shared reward, respectively. A centralized critic that takes as input the observations and actions of all agents are used in both, which makes them hard to scale. PS-TRPO (Gupta et al., 2017) solves problems that were previously considered intractable by most MARL algorithms via sharing of policy parameters that also improves multi-agent cooperation. However, the cooperation is still limited without sharing information among agents. Sharing parameters of value function among agents is considered in (Zhang et al., 2018) and convergence guarantee is provided for linear function approximation. However, the proposed algorithms and their convergence are established only in fully observable environments. Value propagation is proposed in (Qu et al., 2019) for networked MARL, which uses softmax temporal consistency to connect value and policy updates. However, this method only works on networked agents with static connectivity. CommNet (Sukhbaatar et al., 2016) and BiCNet (Peng et al., 2017) communicate the encoding of local observation among agents. ATOC (Jiang & Lu, 2018) and TarMAC (Das et al., 2019) enable agents to learn when to communicate and who to send messages to, respectively, using attention mechanism. These communication models prove that communication does help for cooperation. However, full communication is costly and inefficient, while restrained communication may limit the range of cooperation.

**Graph Convolution and Relation.** Many important real-world applications come in the form of graphs, such as social networks (Kipf & Welling, 2017), protein-interaction networks (Duvenaud et al., 2015), and 3D point cloud (Charles et al., 2017). Several frameworks (Henaff et al., 2015; Niepert et al., 2016; Kipf & Welling, 2017; Velickovic et al., 2017) have been architected to extract locally connected features from arbitrary graphs. A graph convolutional network (GCN) takes as input the feature matrix that summarizes the attributes of each node and outputs a node-level feature matrix. The function is similar to the convolution operation in CNNs, where the kernels are convolved across local regions of the input to produce feature maps. Using GCNs, interaction networks can reason the objects, relations and physics in complex systems, which has been proven difficult for CNNs. A few interaction frameworks have been proposed to predict the future states and underlying properties, such as IN (Battaglia et al., 2016), VIN (Watters et al., 2017), and VAIN (Hoshen, 2017). Relational reinforcement learning (RRL) (Zambaldi et al., 2018) embeds multi-head dot-product attention (Vaswani et al., 2017) as relation block into neural networks to learn pairwise interaction representation of a set of entities in the agent's state, helping the agent solve tasks with complex logic. Relational Forward Models (RFM) (Tacchetti et al., 2019) use supervised learning to predict the actions of all other agents based on global state. However, in partially observable environments, it is hard for RFM to learn to make accurate prediction with only local observation. MAGnet (Malysheva et al., 2018) learns relevance information in the form of a relevance graph, where relation weights are learned by pre-defined loss function based on heuristic rules, but relation weights in DGN are learned by directly minimizing the temporal-difference error of value function end-to-end. Agarwal et al. (2019) used attention mechanism for communication and proposed a

curriculum learning for transferable cooperation. However, these two methods require the objects in the environment are explicitly labeled, which is infeasible in many real-world applications.

## 3 METHOD

We construct the multi-agent environment as a graph, where agents in the environment are represented by the nodes of the graph and each node $i$ has a set of neighbors, $\mathbb{B}_i$, which is determined by distance or other metrics, depending on the environment, and varies over time (*e.g.*, the agents in $i$'s communication range or local observation). Moreover, neighboring nodes can communicate with each other. The intuition behind this is neighboring agents are more likely to interact with and affect each other. In addition, in many multi-agent environments, it may be costly and less helpful to take all other agents into consideration, because receiving a large amount of information requires high bandwidth and incurs high computational complexity, and agents cannot differentiate valuable information from globally shared information (Tan, 1993; Jiang & Lu, 2018). As convolution can gradually increase the receptive field of an agent[1], the scope of cooperation is not restricted. Therefore, it is efficient and effective to consider only neighboring agents. Unlike the static graph considered in GCNs, the graph of multi-agent environment is dynamic and continuously changing over time as agents move or enter/leave the environment. Therefore, DGN should be able to adapt to the dynamics of the graph and learn as the multi-agent environment evolves.

### 3.1 GRAPH CONVOLUTION

The problem is formulated as Decentralized Partially Observable Markov Decision Process (Dec-POMDP), where at each timestep $t$ each agent $i$ receives a local observation $o_i^t$, which is the property of node $i$ in the graph, takes an action $a_i^t$, and gets an individual reward $r_i^t$. The objective is to maximize the sum of all agents' expected returns. DGN consists of three types of modules: observation encoder, convolutional layer and $Q$ network, as illustrated in Figure 1. The local observation $o_i^t$ is encoded into a feature vector $h_i^t$ by MLP for low-dimensional input or CNN for visual input. The convolutional layer integrates the feature vectors in the local region (including node $i$ and its neighbors $\mathbb{B}_i$) and generates the latent feature vector $h_i^{'t}$. By stacking more convolutional layers, the receptive field of an agent gradu-

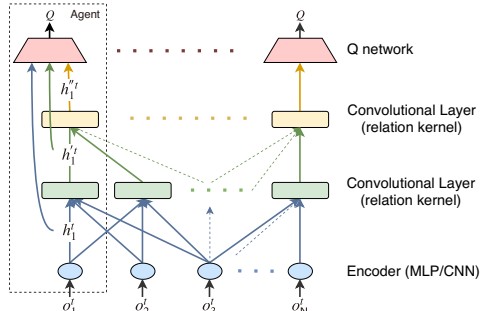

Figure 1: DGN consists of three modules: encoder, convolutional layer, and $Q$ network. All agents share weights and gradients are accumulated to update the weights.

ally grows, where more information is gathered, and thus the scope of cooperation can also increase. That is, by one convolutional layer, node $i$ can directly acquire the feature vectors from the encoders of nodes in one-hop (*i.e.*, $\mathbb{B}_i$). By stacking two layers, node $i$ can get the output of the first convolutional layer of the nodes in one-hop, which contains the information from nodes in two-hop. However, regardless of how many convolutional layers are stacked, node $i$ only communicates with its neighbors. This makes DGN practical in real-world applications, where each agent has limited communication range. In addition, details of the convolution kernel will be discussed in next subsection.

As the number and position of agents vary over time, the underlying graph continuously changes, which brings difficulties to graph convolution. To address the issue, we merge all agents' feature vectors at time $t$ into a feature matrix $F^t$ with size $N \times L$ in the order of index, where $N$ is the number of agents and $L$ is the length of feature vector. Then, we construct an adjacency matrix $C_i^t$ with size $(|\mathbb{B}_i| + 1) \times N$ for agent $i$, where the first row is the one-hot representation of the index of node $i$, and the $j$th row, $j = 2, \ldots, |\mathbb{B}_i| + 1$, is the one-hot representation of the index of the $(j-1)$th neighbor. Then, we can obtain the feature vectors in the local region of node $i$ by $C_i^t \times F^t$.

Inspired by DenseNet (Huang et al., 2017), for each agent, the features of all the preceding layers are concatenated and fed into the $Q$ network, so as to assemble and reuse the observation representation

---

[1]The receptive field of an agent at a convolutional layer is its perceived agents at that layer.

and features from different receptive fields, which respectively have distinctive contributions to the strategy that takes the cooperation at different scopes into consideration.

During training, at each timestep, we store the tuple $(\mathcal{O}, \mathcal{A}, \mathcal{O}', \mathcal{R}, \mathcal{C})$ in the replay buffer, where $\mathcal{O} = \{o_1, \cdots, o_N\}$ is the set of observations, $\mathcal{A} = \{a_1, \cdots, a_N\}$ is the set of actions, $\mathcal{O}' = \{o'_1, \cdots, o'_N\}$ is the set of next observations, $\mathcal{R} = \{r_1, \cdots, r_N\}$ is the set of rewards, and $\mathcal{C} = \{C_1, \cdots, C_N\}$ is the set of adjacency matrix. Note that we drop time $t$ in the notations for simplicity. Then, we sample a random minibatch of size $S$ from the replay buffer and minimize the loss

$$\mathcal{L}(\theta) = \frac{1}{S} \sum_{S} \frac{1}{N} \sum_{i=1}^{N} (y_i - Q(O_{i,\mathcal{C}}, a_i; \theta))^2, \tag{1}$$

where $y_i = r_i + \gamma \max_{a'} Q(O'_{i,\mathcal{C}}, a'_i; \theta')$, $O_{i,\mathcal{C}} \subseteq \mathcal{O}$ denotes the set of observations of the agents in $i$'s receptive fields determined by $\mathcal{C}$, $\gamma$ is the discount factor, and $Q$ function, parameterized by $\theta$, takes $O_{i,\mathcal{C}}$ as input and outputs $Q$ value for agent $i$. The action of agent can change the graph at next timestep. Ideally, $Q$ function should be learned on the changing graph. However, the graph may change quickly, which makes $Q$ network difficult to converge. Thus, we keep $\mathcal{C}$ unchanged in two successive timesteps when computing the $Q$-loss in training to ease this learning difficulty. The gradients of $Q$-loss of all agents are accumulated to update the parameters. Then, we softly update the target network as $\theta' = \beta\theta + (1 - \beta)\theta'$.

Like CommNet (Sukhbaatar et al., 2016), DGN can also be seen as a factorization of a centralized policy that outputs actions for all the agents to optimize the average expected return. The factorization is that all agents share $\theta$ and the model of each agent is connected to its neighbors, dynamically determined by the graph of agents at each timestep. More convolutional layers (*i.e.*, larger receptive field) yield a higher degree of centralization that mitigates non-stationarity. In addition, unlike other methods with parameter-sharing, *e.g.*, DQN, that sample experiences from individual agents, DGN samples experiences based on the graph of agents, not individual agents, and thus takes into consideration the interactions between agents. Nevertheless, the parameter-sharing of DGN does not prevent the emergence of sophisticated cooperative strategies, as we will show in the experiments. Note that during execution each agent only requires the (latent) features from its neighbors (*e.g.*, via communication) regardless of the number of agents, which makes DGN easily scale.

## 3.2 RELATION KERNEL

Convolution kernels integrate the feature in the receptive field to extract the latent feature. One of the most important properties is that the kernel should be independent from the order of the input feature vectors. Mean operation as in CommNet (Sukhbaatar et al., 2016) meets this requirement, but it leads to only marginal performance gain. BiCNet (Peng et al., 2017) uses the learnable kernel, *i.e.*, RNN. However, the input order of feature vectors severely impacts the performance, though the affect is alleviated by bi-direction mechanism. Further, convolution kernels should be able to learn how to abstract the relation between agents so as to integrate their input features.

Inspired by RRL (Zambaldi et al., 2018), we use multi-head dot-product attention as the convolutional kernel to compute interactions between agents. For each agent $i$, let $\mathbb{B}_{+i}$ denote $\mathbb{B}_i$ and $i$. The input feature of each agent is projected to query, key and value representation by each independent attention head. For attention head $m$, the relation between $i$ and $j \in \mathbb{B}_{+i}$ is computed as

$$\alpha_{ij}^m = \frac{\exp\left(\tau \cdot \mathbf{W}_Q^m h_i \cdot (\mathbf{W}_K^m h_j)^\mathsf{T}\right)}{\sum_{k \in \mathbb{B}_{+i}} \exp\left(\tau \cdot \mathbf{W}_Q^m h_i \cdot (\mathbf{W}_K^m h_k)^\mathsf{T}\right)}, \tag{2}$$

where $\tau$ is a scaling factor. For each attention head, the value representations of all the input features are weighted by the relation and summed together. Then, the outputs of $M$ attention heads for agent $i$ are concatenated and then fed into function $\sigma$, *i.e.*, one-layer MLP with ReLU non-linearities, to produce the output of the convolutional layer,

$$h'_i = \sigma(\text{concatenate}[\sum_{j \in \mathbb{B}_{+i}} \alpha_{ij}^m \mathbf{W}_V^m h_j, \forall m \in M]). \tag{3}$$

Figure 2 illustrates the computation of the convolutional layer with relation kernel. Multi-head attention makes the kernel independent from the order of input feature vectors, and allows the kernel

to jointly attend to different representation subspaces. More attention heads give more relation representations and make the training more stable empirically (Vaswani et al., 2017). Moreover, with multiple convolutional layers, higher order relation representations can be extracted, which effectively capture the interplay between agents and greatly help to make cooperative decision.

### 3.3 TEMPORAL RELATION REGULARIZATION

Cooperation is a persistent and long-term process. Who to cooperate with and how to cooperate should be consistent and stable for at least a short period of time even when the state/feature of surrounding agents changes. Thus, the relation representation, *i.e.*, the attention weight distribution over the neighboring agents produced by the relation kernel (Equation 2), should be also consistent and stable for a short period of time. To make the learned

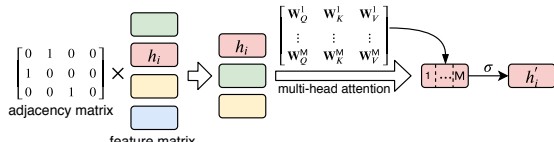

Figure 2: Illustration of computation of the convolutional layer with relation kernel of multi-head attention.

attention weight distribution stable over timesteps, we propose temporal relation regularization. Inspired by temporal-difference learning, we use the attention weight distribution in the next state as the target for the current attention weight distribution. We adopt KL divergence to measure how the current attention weight distribution is different from the target attention weight distribution. Minimizing the KL divergence as a regularization will encourage the agent to form the consistent relation representation and hence consistent cooperation. In CNNs/GCNs, higher layer learns more abstract representation. Similarly, in DGN, the relation representation captured by upper layer should be more abstract and stable. Thus, we apply temporal relation regularization to the upper layer. Moving average and RNN structures might help the relation representation be stable in static graph. However, in dynamic environment where the neighbors of the agent quickly change, averaging or integrating cannot be performed on the attention weights of different neighbors.

It should be noted that we only use target network to produce the target $Q$ value. For the calculation of KL divergence between relation representations in two timesteps, we apply *current* network to the next state to produce the target relation representation. This is because relation representation is highly correlated with the weights of feature extraction. But update of such weights in target network always lags behind that of current network, making the relation representation produced by target network not consistent with that produced by current network.

Let $\mathcal{G}_m^\kappa(O_{i,\mathcal{C}};\theta)$ denotes the attention weight distribution of relation representations of attention head $m$ at convolutional layer $\kappa$ for agent $i$. Then, with temporal relation regularization, the loss is modified as below

$$\mathcal{L}(\theta) = \frac{1}{S}\sum_{S}\frac{1}{N}\sum_{i=1}^{N}((y_i - Q(O_{i,\mathcal{C}}, a_i;\theta))^2 + \lambda\frac{1}{M}\sum_{m=1}^{M}D_{\mathrm{KL}}(\mathcal{G}_m^\kappa(O_{i,\mathcal{C}};\theta)||\mathcal{G}_m^\kappa(O'_{i,\mathcal{C}};\theta)), \quad (4)$$

where $\lambda$ is the coefficient for the regularization loss. Temporal relation regularization of upper layer in DGN helps the agent to form long-term and consistent action policy in the highly dynamical environment with many moving agents. This will further help agents to form cooperative behavior since many cooperative tasks need long-term consistent cooperation among agents to get the final reward. We will further analyze this in the experiments.

## 4 EXPERIMENTS

For the experiments, we adopt a grid-world platform MAgent (Zheng et al., 2017). In the $30 \times 30$ grid-world environment, each agent corresponds to one grid and has a local observation that contains a square view with $11 \times 11$ grids centered at the agent and its own coordinates. The discrete actions are moving or attacking. Two scenarios, *battle* and *jungle*, are considered to investigate the cooperation among agents. Also, we build an environment, *routing*, that simulates routing in packet switching networks. These three scenarios are illustrated in Figure 3. In the experiments, we compare DGN with independent Q-learning, DQN, which is fully decentralized, CommNet (Sukhbaatar et al., 2016), and MeanField Q-learning (MFQ) (Yang et al., 2018b). We also evaluate two variants of DGN for ablation study, which are DGN without temporal relation regularization, denoted

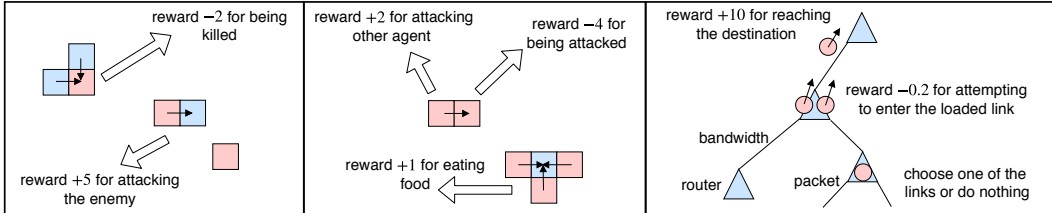

Figure 3: Illustration of experimental scenarios: *battle* (left), *jungle* (mid), and *routing* (right).

as DGN-R, and further DGN-R with mean kernels instead of relation kernels, denoted as DGN-M. In the experiments, DGN and the baselines are *parameter-sharing* and trained using Q-learning. Moreover, to ensure the comparison is fair, their basic hyperparameters are all the same and their parameter sizes are also similar. Please refer to Appendix for hyperparameters and experimental settings. The code of DGN is available at `https://github.com/PKU-AI-Edge/DGN/`.

## 4.1 BATTLE

In this scenario, $N$ agents learn to fight against $L$ enemies who have superior abilities than the agents. The moving or attacking range of the agent is the four neighbor grids, however, the enemy can move to one of twelve nearest grids or attack one of eight neighbor grids. Each agent/enemy has six hit points (*i.e.*, being killed by six attacks). After the death of an agent/enemy, the balance will be easily lost and hence we will add a new agent/enemy at a random location to maintain the balance. By that, we can make fair comparison among different methods in terms of kills, deaths and kill-death ratio besides reward for given timesteps. The pretrained DQN model built-in MAgent takes the role of enemy. As individual enemy is much powerful than individual agent, an agent has to collaborate with others to develop coordinated tactics to fight enemies. Moreover, as the hit point of enemy is six, agents have to consistently cooperate to kill an enemy.

We trained all the models with the setting of $N = 20$ and $L = 12$ for 2000 episodes. Figure 4 shows their learning curves in terms of mean reward. For all the models, the shadowed area is enclosed by the min and max value of three training runs, and the solid line in middle is the mean value (same for jungle and routing). DGN converges to much higher mean reward than other baselines, and its learning curve is more stable. MFQ outperforms CommNet and DQN which first get relative high reward, but eventually converge to much lower reward. As observed in the experiment, at the beginning of training, DQN and CommNet learn sub-optimum policies such as gathering as a group in a corner to avoid being attacked,

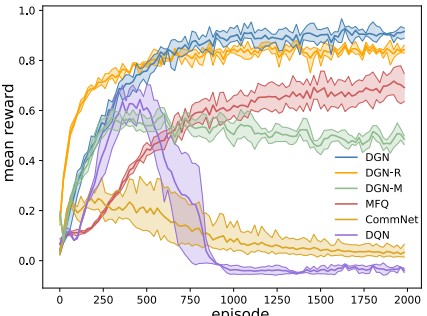

Figure 4: Learning curves in battle.

since such behaviors generate relatively high reward. However, since the distribution of reward is uneven, *i.e.*, agents at the exterior of the group are easily attacked, learning from the "low reward experiences" produced by the sub-optimum policy, DQN and CommNet converge to more passive policies, which lead to much lower reward. We evaluate DGN and the baselines by running 30 test games, each game unrolled with 300 timesteps. Table 1 shows the mean reward, kills, deaths, and kill-death ratio.

DGN agents learn a series of tactical maneuvers, such as encircling and envelopment of a single flank. For single enemy, DGN agents learn to encircle and attack it together. For a group of en-

Table 1: Battle

|  | DGN | DGN-R | DGN-M | MFQ | CommNet | DQN |
|---|---|---|---|---|---|---|
| mean reward | **0.91** | 0.84 | 0.50 | 0.70 | 0.03 | −0.03 |
| # kills | **220** | 208 | 121 | 193 | 7 | 2 |
| # deaths | **97** | 101 | 84 | 92 | 27 | 74 |
| kill-death ratio | **2.27** | 2.06 | 1.44 | 2.09 | 0.26 | 0.03 |

emies, DGN agents learn to move against and attack one of the enemy's open flanks, as depicted in Figure 5a. CommNet agents adopt an active defense strategy. They seldom launch attacks but rather run away or gather together to avoid being attacked. DQN agents driven by self-interest fail to learn a rational policy. They are usually forced into a corner and passively react to the enemy's attack, as shown in Figure 5b. MFQ agents do not effectively cooperate with each other because the mean action incurs the loss of important information that could help cooperation. In DGN, relation kernels can extract high order relations between agents through graph convolution, which can be easily exploited to yield cooperation. Therefore, DGN outperforms other baselines.

**Ablations.** As shown in Figure 4 and Table 1, comparing DGN and DGN-R, we see that the removal of temporal relation regularization incurs slight drop in performance. In the experiment, it is observed that DGN agents indeed behave more consistently and synchronously with each other, while DGN-R agents are more likely to be distracted by the new appearance of enemy or friend nearby and abandon its original intended trajectory. This results in fewer appearances of successful formation of encircling of a moving enemy, which might need consistent cooperation of agents to move across the field. DGN agents often overcome such distraction and show more long-term strategy and aim by moving more synchronously to chase the enemy until encircle and destroy it. From this experiment, we can see that temporal relation regularization indeed helps agents to form more consistent cooperation. Moreover, comparing DGN-R and DGN-M, we confirm that relation kernels that abstract the relation representation between agents indeed helps to learn cooperation. Although DGN-M and CommNet both use mean operation, DGN-M substantially outperforms CommNet. This is attributed to graph convolution can effectively extract latent features from gradually increased receptive field. The performance of DGN with different receptive fields is available in Appendix.

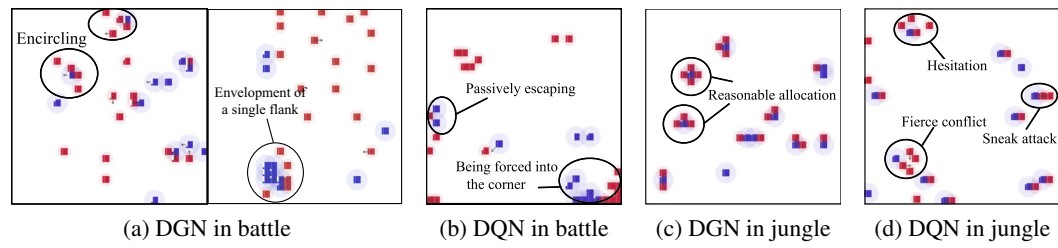

| (a) DGN in battle | (b) DQN in battle | (c) DGN in jungle | (d) DQN in jungle |

Figure 5: Illustration of representative behaviors of DGN and DQN agents in battle and jungle.

## 4.2 JUNGLE

This scenario is a moral dilemma. There are N agents and L foods in the field, where foods are stationary. An agent gets positive reward by eating food, but gets higher reward by attacking other agent. At each timestep, each agent can move to or attack one of four neighboring grids. Attacking a blank grid gets a small negative reward (inhibiting excessive attacks). This experiment is to examine whether agents can learn collaboratively sharing resources rather than attacking each other. We trained all the models in the setting of $N = 20$ and $L = 12$ for 2000 episodes. Table 2 shows the mean reward and number of attacks between agents over 30 test runs, each game unrolled with 120 timesteps. Figure 6 shows their learning

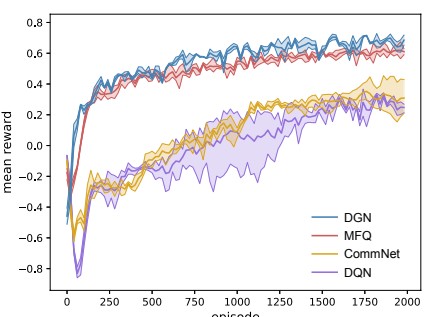

Figure 6: Learning curves in jungle.

curves. DGN outperforms all the baselines during training and test in terms of mean reward and number of attacks between agents. It is observed that DGN agents can properly select the close food and seldom hurt each other, and the food can be allocated rationally by the surrounding agents, as

Table 2: Jungle

|  | DGN | MFQ | CommNet | DQN |
|---|---|---|---|---|
| mean reward | **0.66** | 0.62 | 0.30 | 0.24 |
| # attacks | **1.14** | 2.74 | 5.44 | 7.35 |

Table 3: Routing

| $(N, L)$ | | Floyd | Floyd w/ BL | DGN | MFQ | CommNet | DQN |
|---|---|---|---|---|---|---|---|
| $(20, 20)$ | mean reward | | | **1.23** | 1.02 | 0.49 | 0.18 |
| | delay | 6.3 | 8.7 | **8.0** | 9.4 | 18.6 | 46.7 |
| | throughput | 3.17 | 2.30 | **2.50** | 2.13 | 1.08 | 0.43 |
| $(40, 20)$ | mean reward | | | **0.86** | 0.78 | 0.39 | 0.12 |
| | delay | 6.3 | 13.7 | **9.8** | 11.8 | 23.5 | 83.6 |
| | throughput | 6.34 | 2.91 | **4.08** | 3.39 | 1.70 | 0.49 |
| $(60, 20)$ | mean reward | | | **0.73** | 0.59 | 0.31 | 0.06 |
| | delay | 6.3 | 14.7 | **12.6** | 15.5 | 27.0 | 132.0 |
| | throughput | 9.52 | 4.08 | **4.76** | 3.87 | 2.22 | 0.45 |

shown in Figure 5c. Moreover, attacks between DGN agents are much less than others, *e.g.,* 2× less than MFQ. Sneak attack, fierce conflict, and hesitation are the characteristics of CommNet and DQN agents, as illustrated in Figure 5d, verifying their failure of learning cooperation.

### 4.3 ROUTING

The network consists of $L$ routers. Each router is randomly connected to a constant number of routers (three in the experiment), and the network topology is stationary. There are $N$ data packets with a random size, and each packet is randomly assigned a source and destination router. If there are multiple packets with the sum size larger than the bandwidth of a link, they cannot go through the link simultaneously. In the experiment, data packets are agents, and they aim to quickly reach the destination while avoiding congestion. At each timestep, the observation of a packet is its own attributes (*i.e.*, current location, destination, and data size), the attributes of cables connected to its current location (*i.e.*, load, length), and neighboring data packets (on the connected cable or routers). It takes some timesteps for a data packet to go through a cable, a linear function of the cable length. The action space of a packet is the choices of next hop. Once the data packet arrives at the destination, it leaves the system and another data packet enters the system with random initialization.

We trained all the models with the setting of $N = 20$ and $L = 20$ for 2000 episodes. Figure 7 shows their learning curves. DGN converges to much higher mean reward and more quickly than the baselines. We evaluate all the models by running 10 test games, each game unrolled with 300 timesteps. Table 3 shows the mean reward, mean delay of data packets, and throughput, where the delay of a packet is measured by the timesteps taken from source to destination and the throughput is the number of delivered packets per timestep.

To better interpret the performance of the models, we calculate the shortest path for every pair of nodes in the network using Floyd algorithm. Then, during test, we directly calculate the delay and throughout based on the shortest path of each packet, which is Floyd in Table 3. Note that this delay is without considering the bandwidth limitation (*i.e.*, data packets can go through any link simultaneously). Thus, this is the ideal case for the routing problem. When considering the bandwidth limit, we let each packet follow its shortest path, and if a link is congested, the packet will wait at the router until the link is unblocked. This is Floyd with Bandwidth Limit (BL) in Table 3, which can be considered as the practical solution.

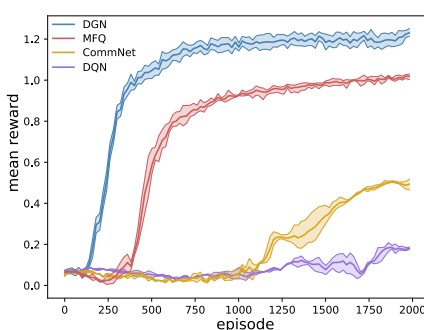

Figure 7: Learning curves in routing.

As shown in Table 3, the performance of DGN is much better than other models and Floyd with BL.

In the experiment, it is observed that DGN agents tend to select the shortest path to the destination, and more interestingly, learn to select different paths when congestion is about to occur. DQN agents cannot learn the shortest path due to myopia and easily cause congestion at some links without considering the influence of other agents. Communication indeed helps as MFQ and CommNet outperform DQN. However, they are unable to develop the sophisticated strategies as DGN does and eventually converge to much lower performance.

To investigate how network traffic affects the performance of the models, we performed the experiments with heavier data traffic, *i.e.*, N = 40 and L = 20, where all the models are directly applied to the setting without retraining. From Table 3, we can see that DGN is much better than Floyd with BL, and MFQ is also better than Floyd with BL. The reason is that Floyd with BL (*i.e.*, simply following the shortest path) is favorable when traffic is light and congestion is rare, while it does not work well when traffic is heavy and congestion easily occurs. We further apply all the models learned in N = 20 and L = 20 to the setting of N = 60 and L = 20. DGN still outperforms Floyd with BL, while MFQ become worse than Floyd with BL. It is observed in the experiments that DGN without retraining outperforms Floyd with BL up to N = 140 and L = 20, available in Appendix. From the experiments, we can see that our model trained with fewer agents can well generalize to the setting with much more agents, which demonstrates that the policy that takes as input the integrated features from neighboring agents based on their relations scales well with the number of agents.

## 5 Conclusions

We have proposed graph convolutional reinforcement learning. DGN adapts to the dynamics of the underlying graph of the multi-agent environment and exploits convolution with relation kernels to extract latent features from gradually increased receptive fields for learning cooperative strategies. Moreover, the relation representation between agents are temporally regularized to make the cooperation more consistent. Empirically, DGN significantly outperforms existing methods in a variety of cooperative multi-agent scenarios.

### Acknowledgments

This work was supported in part by NSF China under grant 61872009, Huawei Noah's Ark Lab, and Peng Cheng Lab.

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

## A HYPERPARAMETERS

Table 4 summarizes the hyperparameters used by DGN and the baselines in the experiments.

Table 4: Hyperparameters

| Hyperparameter | DGN | CommNet | MFQ | DQN |
|---|---|---|---|---|
| discount ($\gamma$) | | $0.96, 0.96, 0.98$ | | |
| batch size | | $10$ | | |
| buffer capacity | | $2 \times 10^5$ | | |
| $\beta$ | | $0.01$ | | |
| $\epsilon$ and decay | | $0.6/0.996$ | | |
| optimizer | | Adam | | |
| learning rate | | $10^{-4}$ | | |
| # neighbors | $3$ | $-$ | $3$ | $-$ |
| # convolutional layers | $2$ | | $-$ | |
| # attention heads | $8$ | | $-$ | |
| $\tau$ | $0.25$ | | $-$ | |
| $\lambda$ | $0.03$ | | $-$ | |
| $\kappa$ | $2$ | | $-$ | |
| # encoder MLP layers | $2$ | $2$ | $-$ | $-$ |
| # encoder MLP units | $(512, 128)$ | $(512, 128)$ | $-$ | $-$ |
| $Q$ network | affine transformation | affine transformation | $(1024, 256)$ | $(1024, 256)$ |
| MLP activation | | ReLU | | |
| initializer | | random normal | | |

## B EXPERIMENTAL SETTINGS

In jungle, the reward is $0$ for moving, $+1$ for attacking (eating) the food, $+2$ for attacking other agent, $-4$ for being attacked, and $-0.01$ for attacking a blank grid. In battle, the reward is $+5$ for attacking the enemy, $-2$ for being killed, and $-0.01$ for attacking a blank grid. In routing, the bandwidth of each link is the same and set to $1$. Each data packet is with a random size between $0$ and $1$. If the link to the next hop selected by a data packet is overloaded, the data packet will stay at the current router and be punished with a reward $-0.2$. Once the data packet arrives at the destination, it leaves the system and gets a reward $+10$. In the experiments, we fix the size of $\mathbb{B}$ to 3, because DGN is currently implemented based on TensorFlow which does not support dynamic computing graph (varying size of $\mathbb{B}$). We also show how different sizes of $\mathbb{B}$ affect DGN's performance in the following. Indeed, DGN adapts to dynamic environments, no matter how the number of agents changes, how the graph of agents changes, and how many neighbors each agent has.

## C ADDITIONAL EXPERIMENTS

As aforementioned, larger receptive field yields a higher degree of centralization that mitigates non-stationarity. We also investigate this in the experiments. First we examine how DGN performs with different number of convolution layers. As illustrated in Figure 8, two convolutional layers indeed yield more stable learning curve than one layer as expected.

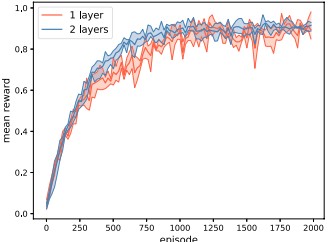

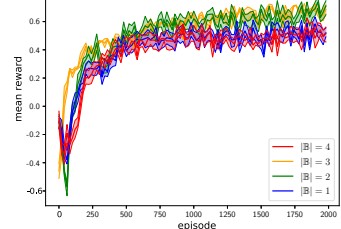

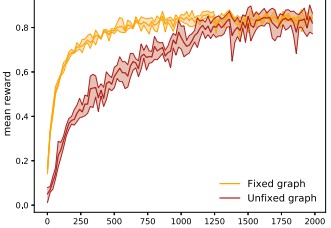

Figure 8: DGN with different number of convolutional layers in battle.

Figure 9: DGN with different number of neighbors for each agent in jungle.

Figure 10: Learning curves of DGN on fixed graph and unfixed graph in battle.

We also investigate how the size of neighbors $|\mathbb{B}|$ affects the performance of DGN. We set $|\mathbb{B}|$ of each agent to 1, 2, 3 and 4 in jungle. As illustrated in Figure 9, when $|\mathbb{B}|$ increases from 1 to 3, the performance improves. However, when $|\mathbb{B}| = 4$, the performance drops, equivalent to $|\mathbb{B}| = 1$. In addition, as shown in Figure 6, the full communication method, CommNet, has very limited performance. These verify that it may be less helpful and even negatively affect the performance to take all other agents into consideration.

Ideally, $Q$ function should be learned on the changing graph of agents. However, the quickly changing graph can make $Q$ function difficult to converge. Thus, we fix the graph in two successive timesteps to mitigate the effect of changing graph and ease the learning difficulty. As shown in Figure 10, the learning curve of fixed graph indeed converges faster than that of unfixed graph. As keeping the graph of agents unchanged is necessary for temporal relation regularization, for fair comparison, we also remove temporal relation regularization for fixed graph.

We also perform additional experiments to compare DGN with ATOC and TarMAC in Battle. As shown in Figure 11, DGN outperforms ATOC. The reason is that LSTM kernel is worse than multi-head attention kernel in capturing relation between agents. Like CommNet, TarMAC is also a full communication method. Similarly, DGN also outperforms TarMAC. This again verifies that receiving redundant information may negatively affect the performance.

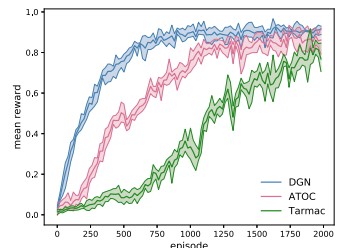

Figure 11: Learning curves of DGN, ATOC and TarMAC in battle.

Figure 12: DGN versus Floyd with BL under increasingly heavier traffic in routing.

We also conducted additional experiments in routing to compare DGN (learned in the setting of $N = 20$ and $L = 20$) and Floyd with BL under increasingly heavier traffic, in terms of mean delay. As shown in Figure 12, DGN continuously outperforms Floyd with BL up to $N = 140$. After that, Floyd with BL outperforms DGN. The reason is that when the traffic becomes so heavy, the network is fully congested and there is no way to improve the performance. DGN learned in much lighter traffic may still try to find better routes, but this incurs extra delay.

Figure 13 and Figure 14 show the learning curves of DGN, DGN-R and DGN-M in jungle and routing, respectively. We can see that DGN and DGN-R outperforms DGN-M. DGN slightly outperforms DGN-R in routing, and they performs similarly in jungle. The reason is that in both jungle and routing the agents do not require cooperation consistency as much as in battle. In battle the agents need to cooperatively and consistently attack an enemy since it has six hit points. However, in jungle agents seldom move once they reach the status of sharing food, while in routing data packets (agents) with different destinations seldom share many links (cooperate continuously) along their paths.

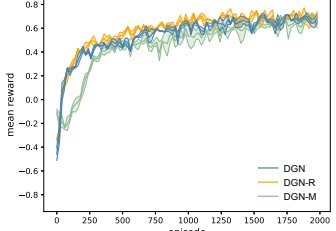

Figure 13: Learning curves of DGN, DGN-R, and DGN-M in jungle.

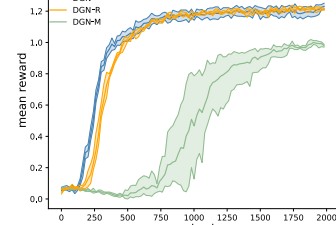

Figure 14: Learning curves of DGN, DGN-R, and DGN-M in routing.

