# OpenReview forum: "Graph Convolutional Reinforcement Learning"
_ICLR.cc/2020/Conference — Accept (Poster)_

### Official Review · AnonReviewer3 · 2019-10-07
**Official Blind Review #3**

**Rating:** 6

**Review:**

This paper introduces Graph Convolutional Reinforcement Learning (referred to as DGN). DGN is a Deep Q-Learning (DQN) agent structured as a graph neural network / graph convolutional network with multi-head dot product attention as a message aggregation function. Graphs are obtained based on spatial neighborhoods (e.g. k nearest neighbors) or based on network structure in the domain. DGN considers a multi-agent setting with a centralized learning algorithm and shared parameters across all (controlled) agents, but individually allocated reward. Further, the paper considers environments where other non-learning agents are present which follow a pre-trained, stationary policy. In addition to the attention-based multi-agent architecture, the paper introduces a regularizer on attention weights similar to the use of target networks in DQN, to stabilize training. Results demonstrate that the proposed model architecture outperforms related earlier agent architectures that do not use attention or use a fully-connected graph.

Overall, this paper addresses an interesting problem, introduces a novel combination of well-established architecture/agent building blocks and introduces a novel regularizer. The novelty and significance of the contributions, however is limited, as many recent works have explored using graph-structured representations and attention in multi-agent domains (e.g. VAIN: Hoshen (NeurIPS 2017), Zambaldi et al. (ICLR 2019), Tacchetti et al. (ICLR 2019)). The combination of these blocks and the considered problem setting is novel, but otherwise incremental. Nonetheless, the results are interesting, the overall architecture is simple (which I consider to be a good sign), and the attention regularizer is novel, hence I would rate this paper as relevant to the ICLR audience.

My main concern with this paper is clarity of writing: I have the feeling that important details are missing and some modeling decisions and formulas are difficult to understand. For example, I found section 3.3 difficult to read. The following sentences/statements need revision or further explanation:
* “Intuitively, if the relation representation produced by the relation kernel of upper layer truly captures the abstract relation between surrounding agents and itself, such relation representation should be stable/consistent” (Please clarify)
* “We use the attention weight distribution in the next state as the target for the current attention weight distribution” (What is the reasoning behind this? Would an exponential moving average of attention logits/weights work as well?)
* “While RNN/LSTM forces consistent action, regardless of cooperation” (unclear)
* “Since we only focus on the self-consistent of the relation representation based on the current feature extraction network we apply current network to the next state to produce the new relation representation instead of the target network as in deep Q learning” (unclear)
* The KL term in Eq. 4 is odd: z_i is defined as G^K and vice versa, neither of them appear to be distributions. I suppose one of the two arguments of the KL term should be the attention distribution for the current time step and the other argument for the next time step (if I understood the motivation in the earlier paragraph correctly), but this is not evident from Eq. 4.
* KL is not symmetric -- what motivates the particular ordering in your case? Did you consider symmetric divergences such as Jensen-Shannon divergence (JSD)?

I also wonder about the necessity of assembling adjacency matrices per node to create an intermediate ordered representation of the neighborhood on which, afterwards, an order-invariant operation such as mean pooling or self-attentive pooling is applied. Wouldn't it be more efficient to implement these operations directly using sparse scatter/gather operations as most recent GNN frameworks implement these techniques (e.g. PyTorch Geometric or DeepMind's graph_nets library)?

Further, important experimental details are missing, e.g., how observations / node features are represented / obtained from the environment and preprocessed. Do you encode position (continuous/discrete) and normalize in some way? It should further be mentioned that some of the baselines are trained with a different training algorithm and do not only differ in agent architecture (e.g. CommNet) — what is the effect of this?

Experimentally, the results seem sound, but the variance in the results is suprisingly low (see e.g. Figure 7 DQN) — did you change the random seed between runs (both environment seed and the seed for initializing the agent weights)?

Overall, this paper is interesting but needs revision in terms of clarity. Novelty is incremental, but if the paper would otherwise be very well written, I think it could qualify for acceptance. In its current state, I recommend a weak reject.


--- UPDATE AFTER REVISION ---
The clarity in the revised manuscript is significantly improved and I feel confident in recommending acceptance of the paper.

**Experience Assessment:**

I have read many papers in this area.

**Review Assessment: Checking Correctness Of Derivations And Theory:**

I assessed the sensibility of the derivations and theory.

**Review Assessment: Checking Correctness Of Experiments:**

I assessed the sensibility of the experiments.

**Review Assessment: Thoroughness In Paper Reading:**

I made a quick assessment of this paper.

---

> ### Author Response · Authors · 2019-11-14
> **Responses to Review #3**
>
>
>
> We have rewritten the section of temporal relation regularization to better present our idea. The first two paragraphs of Section 3.3 have addressed the reviewer’s comments on temporal relation regularization. In the following, we first list the responses to these comments for reference, and then give the responses to other comments.
>
> >>> ‘Intuitively, if the relation representation produced by the relation kernel of upper layer truly captures the abstract relation between surrounding agents and itself, such relation representation should be stable/consistent’ (Please clarify)
>
> In CNNs/GCNs, higher layer learns more abstract representation. Similarly, in DGN, the relation representation captured by upper layer should be more abstract and stable. This is the motivation of applying temporal relation regularization to the upper layer.
>
> >>> ‘We use the attention weight distribution in the next state as the target for the current attention weight distribution’ (What is the reasoning behind this? Would an exponential moving average of attention logits/weights work as well?)
>
> Cooperation is a persistent and long-term process. Who to cooperate with and how to cooperate should be consistent and stable for at least a short period of time even when the state/feature of surrounding agents changes. Thus, the attention weight distribution over the neighboring agents should be also consistent and stable for a short period of time. To make the learned attention weight distribution stable over timesteps, we use the attention weight distribution in the next state as the target for the current attention weight distribution.
>
> In dynamic environment where the neighbors of agent quickly change, moving average and RNN structures cannot be performed on the attention weights of different neighbors and thus do not work.
>
> >>> ‘Since we only focus on the self-consistent of the relation representation based on the current feature extraction network we apply current network to the next state to produce the new relation representation instead of the target network as in deep Q learning’ (unclear)
>
> For the calculation of KL divergence between relation representations in two timesteps, we apply current network to the next state to produce the target relation representation. This is because relation representation is highly correlated with the weights of feature extraction. But update of such weights in target network always lags behind that of current network, making the relation representation produced by target network not consistent with that produced by current network.
>
> >>> Explaining the arguments in the KL term in Eq. 4.
>
> You understood it correctly. But we need to point out that $\mathcal{G}_m^{\kappa}$, the attention weights after the softmax (Equation 2), are actually a distribution over its neighbors, with the probability $\alpha_{ij}$ for each neighbor $j$. Thus, we could use KL divergence to measure the difference.
>
> >>> “KL is not symmetric -- what motivates the particular ordering in your case? Did you consider symmetric divergences such as JSD?”
>
> Symmetry is not necessary. We use the attention weights in the next state as the target and only update the attention weights at the current state to make it close to the target. KL(current|target) measures how the current attention weight distribution is different from the target attention weight distribution. We also tested MSE, a symmetric metric, and it also works. However, the performance of KL divergence is better.
>
> >>>Assembling adjacency matrices
>
> Assembling adjacency matrices is not necessary but an easy way to implement multi-head attention. This technique is also used in GAT (ICLR 2018) (keras version). In fact, our implementation is also efficient and highly parallel on GPUs, since the computation of the algorithm is realized by dot product. GNN libraries might help the efficiency and are compatible with DGN. We will try these techniques to investigate the difference.
>
> >>> “It should further be mentioned that some of the baselines are trained with a different training algorithm and do not only differ in agent architecture (e.g. CommNet) — what is the effect of this?”
>
> All the baselines are trained with Q-learning and only differ in agent architecture. As pointed out in the CommNet paper, CommNet can be combined with standard RL algorithms or supervised learning. We use the Q-learning version as our baseline.
>
> >>> About observation and preprocessing
>
> In the experiments that the observation of each agent is a square view with 11 × 11 grids centered at the agent and its own coordinates, which is provided by MAgent without preprocessing.
>
> >>> About the random seeds
>
> We indeed changed the random seeds for each run. In routing, each router connects other three routers, and thus the action space of each agent (packet) is small. Therefore, the convergence process of each algorithm is prone to be similar under different random seeds.

---

> > ### Comment · AnonReviewer3 · 2019-11-14
> > **Response**
> >
> > Thank you for your clarifications and for updating the manuscript. The clarity in the revised manuscript is indeed significantly improved and I feel confident in recommending acceptance of the paper.
> >
> > One minor comment: it might be useful to run the paper through a spelling/grammar checker (an automated service should suffice) to fix some small grammar mistakes, such as "in dynamic environment where the neighbors of agent quickly change".

---

### Official Review · AnonReviewer1 · 2019-10-24
**Official Blind Review #1**

**Rating:** 6

**Review:**

This paper proposes an algorithm allowing "cooperation" between agents in multi-agent reinforcement learning, modeling agents as nodes in a graph. Each agent having only a partial view of the environment, the proposed algorithm uses multi-head attention as a (graph) convolution kernel but otherwise remains similar to the DQN algorithm. Performance is evaluated on three tasks using the MAgent framework.

The paper is reasonably well motivated, grounded and written. It addresses an interesting question: how to make agents cooperate in an efficient way? It does so by combining ideas from two lines of work, bringing incremental novelty.

My main concern relates to the experiments. It seems that ATOC and TarMAC would be the best baselines to compare against for a fair evaluation of the algorithm. Could they be added?

One question for the authors: at the beginning of Section 3, it is stated that "it may be costly and less helpful to take all other agents into consideration". It seems counter intuitive that DGN with several convolutional layers (to have a large receptive field) would be less costly than directly receiving global information? And isn't, in a sense, DGN also making use of global information when it has a large enough receptive field, even if indirectly? In this case, would it also make sense to more thoroughly compare DGN with RFM or other global state algorithms? Can this be clarified?

Finally, readability is somewhat hindered by several small issues:
- Acronyms used in the paper should really be introduced, at least when they are first used. DGN is never introduced, DGN-R/DGN-M are introduced several paragraphs after being first mentioned and BL needs some guessing.
- Re-citing the same paper several times when mentioned in different sections is good practice. I found myself going over and over back to the related work section to find references and acronyms.

Some typos:
Page 1: among all agent -> among all agents
Page 3: of S -> of size S
Page 4: weighed -> weighted, concate -> concatenate
Page 5: respecitvely -> respectively
Page 6: regularation


**Experience Assessment:**

I do not know much about this area.

**Review Assessment: Checking Correctness Of Derivations And Theory:**

N/A

**Review Assessment: Checking Correctness Of Experiments:**

I assessed the sensibility of the experiments.

**Review Assessment: Thoroughness In Paper Reading:**

I read the paper at least twice and used my best judgement in assessing the paper.

---

> ### Author Response · Authors · 2019-11-14
> **Responses to Review #1**
>
>
>
> DGN and all the baselines we compared with are Q-learning algorithms, but ATOC and TarMAC are actor-critic algorithms. Moreover, TarMAC uses a centralized critic that optimizes the shared global reward, which is different from other methods. Thus, they are not quite suitable for fair comparison. Nevertheless, we performed additional experiments to compare with them. As shown in Figure 11, DGN outperforms ATOC. The reason is that LSTM kernel is worse than multi-head attention kernel in capturing relation between agents. Like CommNet, TarMAC is also a full communication method. Similarly, DGN also outperforms TarMAC. The reason is that receiving redundant information may negatively affect the performance.
>
> We consider the case where the state information is unavailable and the agent can only use observations/encodings from other agents to learn to construct and exploit more centralized information. This is true in many real-world applications. However, collecting more observations/encodings incurs more costs and irrelevant observations/encodings can even negatively affect the performance. Because of this, as shown in Figure 9, the performance of $|\mathbb{B}|$=4 is worse than $|\mathbb{B}|$=3 or 2. RFM requires the true state information for supervise learning, and thus RFM is not suitable for fair comparison.
>
> Moreover, we have revised the paper to properly use the acronyms, cite the references, and fix the typos.

---

### Official Review · AnonReviewer4 · 2019-10-31
**Official Blind Review #4**

**Rating:** 6

**Review:**

The paper addresses the problem of coordination in the multi-agent reinforcement learning setting. It proposes the value function factorization similar to independent Q-learning conditioning on the output of the graph convolutional neural network, where the graph topology is based on the agents’ nearest neighbours. The paper is interesting and has some great ideas, for example, KL term to ensure temporal cooperation consistency. However, the paper has its drawbacks and I feel obliged to point them out below. I vote for the weak acceptance of this paper.

One of the main drawbacks of the paper is that it is extremely hard to grasp. Even the Abstract and Introduction are hard to understand without having a pass over the whole paper. The authors often use vague terms such as 'highly dynamic environments' or 'dynamics of the graph' which make it hard to understand what they mean. The paper would benefit from a more precise language. Some of the important notions of the paper are used before they are introduced, which make the general picture very hard to understand and to relate the work to the existing research.

'Related Work' section seems to be missing some recent work applying graph neural networks to multi-agent learning settings:
• Malysheva, Aleksandra, Tegg Taekyong Sung, Chae-Bong Sohn, Daniel Kudenko, and Aleksei Shpilman. "Deep Multi-Agent Reinforcement Learning with Relevance Graphs." arXiv preprint arXiv:1811.12557 (2018).
• Agarwal, Akshat, Sumit Kumar, and Katia Sycara. "Learning Transferable Cooperative Behavior in Multi-Agent Teams." arXiv preprint arXiv:1906.01202 (2019).

My questions to the authors:

• In section 3 you mention 'a set of neighbours ..., which is determined by distance or other metrics'. Can you elaborate on that? What are these metrics in your case?
• Just before the Section 3.1, you say 'In addition, in many multi-agent environments, it may be costly and less helpful to take all other agents into consideration.' Have you run any experiments on that? In the appendix, you show, that making the neighbourhood smaller negatively affects the performance, but what if you make it bigger? Ideally, I would like to see an extended version of Figure 8 and 9 in the main part of the paper since they are very interesting and important for the claims the paper makes.
• At the end of Section 3.1, you mention the soft update of the target network. Later, in 3.3, you say that that you do not use the target network. Can you elaborate more on that?
• In Equation 4, is it a separate KL for each of the attention heads? If yes, this is not clear from the formula.
• It will be useful to see the ablation experiments for all of the testbeds, not only for Battle.
• Why do you think the DQN performance drops in the second half of the training in Figure 4 for all of the runs?
• Have you tried summation instead of the mean aggregation step?

I will put comments for particular parts of the paper below.

ABSTRACT

>>> ...environments are highly dynamic

What do you mean precisely here?

>>> ...graph convolution adapts to the dynamics of the underlying graph of the multi-agent environment

What is the 'dynamics of the underlying graph'? What is the 'graph of the multi-agent environment'?

>>> 'coordination is further boosted'

Not sure that 'boosted' is the right word here.

INTRODUCTION

>>> '...where mutual interplay between humans is abstracted by their relations'

Not sure what it means.

>>> we consider the underlying graph of agents...

The agent graph has not been introduced yet.

>>> DGN shares weights among all agent(s) making it easy to scale

What do you mean precisely by 'easy to scale'? Can you support this claim?

>>> We empirically show the learning effectiveness of DGN in jungle

Needs a reference to the testbed.

>>>  ... interplay between agents and abstract relation representation

What is 'abstract relation representation?

>>> We consider partially observable environments.

What do you mean precisely by that? What is the MDP formalism most suitable for your problem statement? What is objective under your formalism?

>>> However, more convolutional layers will not increase the local region of node i.

What do you mean by that?

>>> As the number and position of agents vary over time, the underlying graph continuously changes, which brings difficulties to graph convolution.

What kind of difficulties?

>>> As the action of agent can change the graph at next timestep which makes it hard to learn Q function.

Why does it make it hard?

>>> DGN can also be seen as a factorization of a centralized policy that outputs actions for all the agents to optimize the average expected return.

It would be useful for the reader to compare your approach with all the others type of the value function factorization. To me, your approach looks like a more sophisticated version of independent Q-learning, is that true?

Minor comments:

* In 3.2 it would be very helpful to put the dimensions for all of the variables for easier understanding.
* The brackets in the equation 3 are not very clear (what are you concatenating across?)
* In section 4, when describing an environment you say 'local observation that contains a square view with 11x11 grids'. What is the total size of the environment?
* The performance plots for Battle include ablations before the ablation subsection is introduced. This is a bit confusing.
* All figures/tables captions should be more detailed and descriptive.
* ‘However, causal influence is not directly related to the reward of environment.’ Should be ‘of the environment’.



**Experience Assessment:**

I have read many papers in this area.

**Review Assessment: Checking Correctness Of Derivations And Theory:**

N/A

**Review Assessment: Checking Correctness Of Experiments:**

I carefully checked the experiments.

**Review Assessment: Thoroughness In Paper Reading:**

I read the paper thoroughly.

---

> ### Author Response · Authors · 2019-11-14
> **Responses to Review #4: part 2**
>
>
>
> >>> About the dynamic graph
>
> We model the multi-agent environment as a graph, where each agent is a node and there is an edge between an agent and each neighbor. As agents keep moving and their neighbors changes quickly, the graph is highly dynamic. We have made this clear in the revision.
>
> >>> “What do you mean precisely by easy to scale? Can you support this claim?”
>
> Since all agents use the same neural network weights, we can directly apply the models trained with small-scale agents to the large-scale scenario. In routing, we apply the trained models (N=20) to the setting from N=40 to N=200. As illustrated in Table 3 and Figure 12, DGN continuously outperforms Floyd with BL up to N = 140.
>
> >>> About the testbed jungle
>
> Jungle is a testbed designed by ourselves. It is a typical social dilemma where agents must learn to eat foods together and avoid to attack each other.
>
> >>> MDP formalism in partially observable environments.
>
> Our problem is a POMDP, where each agent gets a partial observation of the state and obtains a local reward. The objective is to maximize the sum of all agents’ expected returns.
>
>
> >>> The meaning of ‘more convolutional layers will not increase the local region of node i.’
>
> We mean regardless of how many convolutional layers are stacked, node i only communicates with its neighbors. This makes DGN practical in real-world applications, where each agent has limited communication range (e.g., wireless communication).
>
>
> >>> The difficulty of learning on the changing graph of agents.
>
> Ideally, Q-function should be learned on the changing graph of agents. However, the graph changes quickly, which makes Q-function difficult to converge. Fixing the graph in two successive timesteps mitigates the effect of changing graph and eases the learning difficulty. We performed additional experiments to investigate this. Figure 10 shows that fixing the graph indeed speed up the learning. Moreover, keeping the agent graph unchanged is also necessary for temporal relation regularization.
>
> >>> Explaining the factorization by DGN.
>
> The objective is to optimize the sum of all agents’ expected returns. DGN factorizes the problem by each agent optimizes its own local reward, similar to CommNet, BiCNet, etc. Note that this factorization is different from VDN, QMIX and QTRAN where all agents share a global environmental reward and there is still a centralized Q-function that directly optimizes the shared reward during training. In DGN, CommNet and BiCNet, as each agent learns to optimize its own local reward, you could also see them as sophisticated independent Q-learning.
>
>
> Moreover, in Equation 3, we concatenate the output of each attention head, which is described in the paragraph above the equation. The size of the environment is 30x30.

---

> > ### Comment · AnonReviewer4 · 2019-11-15
> > **Reviewer response**
> >
> > >>>  ...This makes it hard to learn abstract representations of mutual interplay between agents.
> >
> > I don't understand why it is the case.
> >
> > >>> Our problem is a POMDP, where each agent gets a partial observation of the state and obtains a local reward. The objective is to maximize the sum of all agents’ expected returns.
> >
> > It would be beneficial for the paper and all its readers if you write down the precise formalism. Is it Dec-POMDP?
> >
> > >>> However, the graph changes quickly, which makes Q-function difficult to converge.
> >
> > I still don't understand why it is the case. Thanks for the additional experiments, however, I would like to see similar experiments for the other two domains.

---

> > > ### Author Response · Authors · 2019-11-15
> > > **Response**
> > >
> > >
> > >
> > > >>>  ...This makes it hard to learn abstract representations of mutual interplay between agents.
> > >
> > > First, the mutual interplay between agents is hard to be quantitatively represented. Moreover, multi-agent environments are changing quickly, which is a result caused by all agents, so it is hard to capture the pairwise relation. Our relation kernel is a neat method to quantitatively represent the pairwise relation.
> > >
> > > >>> "It would be beneficial for the paper and all its readers if you write down the precise formalism. Is it Dec-POMDP?"
> > >
> > > Yes, it is Dec-POMDP. We have added the formalism at the first paragraph of Section 3.1.
> > >
> > > >>> "I still don't understand why it is the case. Thanks for the additional experiments, however, I would like to see similar experiments for the other two domains."
> > >
> > > The graph of the agents changes quickly. The change of the graph at next state will cause the change of target Q value. This is a problem of moving target, which is similar to the problem the target network addressed in DQN. That is the reason why  Q-function is difficult to converge. We really do not have enough time to perform additional experiments on other two scenarios before the deadline of rebuttal.

---

> ### Author Response · Authors · 2019-11-14
> **Responses to Review #4: part 1**
>
>
>
> >>> Related work
>
> Thanks for bringing up the missing references. MAGnet [Malysheva et al., 2018] learns relevance information in the form of a relevance graph, where the relation weights are learned by pre-defined loss function based on heuristic rules, but relation weights in DGN are learned by directly minimizing the temporal-difference error of value function end-to-end. Agarwal et al. (2019) used attention mechanism for communication and proposed a curriculum learning for transferable cooperation. However, these two methods require the objects in the environment are explicitly labeled, which is infeasible in many real-world applications. However, DGN agents only use their raw local observation. We have included these references and clarified the differences in the reversion.
>
> >>> The metrics to determine the neighbor set.
>
> The set of neighbors of an agent could be the agents in its local observation, or the agents within its communication range. It depends on specific scenarios. In the experiments, we use distance and select k-nearest agents as the neighbors and we have also investigated how the number of neighbors affects the performance.
>
> >>> Additional experiment to verify the claim that it may be costly and less helpful to take all other agents into consideration.
>
> Thanks for your constructive suggestion. We have performed additional experiments on bigger neighborhood. As shown in Figure 9, when we set $|\mathbb{B}|$ = 4, the performance drops. In addition, as shown in Figure 6, the full communication method, CommNet, has very limited performance. These verify that it may be less helpful and even negatively affect the performance to take all other agents into consideration. Due to limited time, we have not yet reconstructed the paper to incorporate Figure 8 and 9 in the main part of the paper. We will do that in the final version.
>
> >>> “At the end of Section 3.1, you mention the soft update of the target network. Later, in 3.3, you say that that you do not use the target network. Can you elaborate more on that?”
>
> We indeed use the target network to produce the target value for computing TD-error of Q function. However, for the calculation of the KL divergence between relation representations in two timesteps, we use current network instead of target network. The reason is explained in detail in the second paragraph of Section 3.3.
>
> >>> “In Equation 4, is it a separate KL for each of the attention heads? If yes, this is not clear from the formula.”
>
> Yes, it is the separate KL for each of the attention heads, we have made this clear in Equation 4 in the reversion.
>
> >>> Explaining the ablation experiments for other testbeds.
>
> In other two scenarios, the same conclusions can also be drawn by ablation, but not as significant as in battle. Thus, we neglect the ablation results in these two scenarios for clarity.
>
> >>> “Why does the DQN performance drop in the second half of the training in Figure 4 for all of the runs?”
>
> The enemy model built in MAgent is very powerful, making the Battle game difficult. We watched and analyzed their behaviors for all the runs. As described in Section 4.1, at the beginning, DQN agents learn sub-optimum strategies such as gathering at a corner to avoid to be attacked. These strategies might help at the beginning, and thus the reward is relatively high. But the agents at the edge of the group are easily attacked, receiving low reward and making the reward unevenly distributed among the group. Fitting the 'low reward data' produced by the sub-optimum policy, the DQN converges to more passive policy, e.g., moving disorderly. That is the reason that the mean reward decreases in the later phase.

---

> > ### Comment · AnonReviewer4 · 2019-11-15
> > **Reviewer's response**
> >
> > I appreciate the time and effort the authors invested in improving their paper.
> >
> > >>> However, these two methods require the objects in the environment are explicitly labeled, which is infeasible in many real-world applications.
> >
> > Can you, please, clarify this?
> >
> > >>> We have performed additional experiments on bigger neighborhood.
> >
> > Do I understand correctly, that the total number of agents was 20 there? If yes, it would be interesting to increase the neighbourhood even more.
> >
> > >>> In other two scenarios, the same conclusions can also be drawn by ablation, but not as significant as in battle. Thus, we neglect the ablation results in these two scenarios for clarity.
> >
> > I believe that if you make a claim which is not supported by the other two experiments, then the claim might be wrong. In this case, removing the ablation results does not add clarity, but hides the important information.

---

> > > ### Author Response · Authors · 2019-11-15
> > > **Response**
> > >
> > >
> > >
> > > >>> Clarifying ‘these two methods require the objects in the environment are explicitly labeled, which is infeasible in many real-world applications.’
> > >
> > > These two methods use the entities in the environment as the nodes of the graph. So, they need explicitly know what the entities are and where they are in the environment to construct the graph at each timestep. However, in real-world applications, such information cannot be obtained.
> > >
> > > >>> About increasing the neighbourhood even more.
> > >
> > > When $|\mathbb{B}|$=4, the receptive field of the second convolutional layer is 1+4*(1+4)=21. It is able to cover all of the 20 agents. And the experiments of $|\mathbb{B}|$=1,2,3, and 4 have verified our claims about how the size of neighbors $|\mathbb{B}|$ affects the performance of DGN. When increasing the neighborhood even more, the method will become a full communication method.
> > >
> > > >>> "I believe that if you make a claim which is not supported by the other two experiments, then the claim might be wrong. In this case, removing the ablation results does not add clarity, but hides the important information."
> > >
> > > In the revise version, we have added the ablation study of DGN in jungle and routing in Appendix. Please refer to the last paragraph of Appendix and Figure 15 and 16 for details.

---

### Public Comment · ~Hopeful_Rational2 · 2019-10-07
**Use of KL divergence**

Hi. The paper is nice. However, could you please elaborate more on the use of KL divergence for computing the distance between the attention weight distributions. Thanks.

---

> ### Author Response · Authors · 2019-10-07
> **Re: Use of KL divergence**
>
> After the softmax, the attention weight in the local region is a distribution, with the probability $\alpha_{ij}$. We use KL divergence to measure the difference between the attention weight distributions in two timesteps. Please refer to the code for more details.

---

> ### Public Comment · ~Hopeful_Rational2 · 2019-10-07
> **KL divergence**
>
> Thanks for the quick reply. Is there any specific reason for using KL divergence instead of any other divergence measures?

---

> > ### Author Response · Authors · 2019-10-07
> > **RE:KL divergence**
> >
> > We tested MSE and it also works. However, the performance of KL divergence is better. As mentioned in the paper, the relation in different timesteps should not be the same but similar, thus we use KL divergence to compute the distance between the distributions.

---

### Public Comment · ~Huiknight_Li1 · 2019-10-25
**The released code lacks some key files.**

I am interested in your work and try to reproduce your work. However, there is no any comments about in the released code and the running environment.  Could you please update the code to make it easily be reproduced ?

---

> ### Author Response · Authors · 2019-10-28
> **RE: The released code lacks some key files.**
>
> There is a readme file in Battle and Jungle, respectively. Please check if it helps. Let us know if it does not.

---

### Author Response · Authors · 2019-11-14
**To all the reviewers**

We appreciate the efforts made by the anonymous reviewers on reviewing our paper. Many thanks for the comments which are especially useful for us to improve the quality of this paper. In this revised version, we have carefully addressed the concerns of the reviewers by fixing the problems, performing additional experiments, rewriting the section of temporal relation regularization, and adding necessary references and explanations. We have also improved the writing of the paper. We hope that the reviewers will find our revision satisfactory.

---

### Public Comment · ~Douglas_De_Rizzo_Meneghetti1 · 2020-02-11
**Questions related to the graph and some spelling/notation mistakes**

The paper mentions the creation of a matrix $F^t$ which is invariant to the ordering of nodes in the graph, but what happens in the experiments if one of the agents (or enemies) die? Are all adjacency values for all other agents zeroed for the entity that died? Conversely, what happens if a new agent is added to the environment? Can all the matrices and the overall model accommodate this?

I missed a figure exemplifying how graphs are created, how they change over time or even what they look like. Are all nodes in the graph agents in the same team or does the graph model other things, such as adversaries and environment objects? Why are the edges determined by an arbitrary distance measure if the messaging between nodes is later weighted by self-attention? Couldn't the graph be complete and the attention weights learned to allow the model to learn what to ignore?

Problem with notation: In page 3, L is used as "the length of feature vector". In page 5, L is used as number of enemies.

The acronym "DGN" is never defined. I suppose it was chosen to establish the model as a graph-convolutional variant of DQN, but it would be nice to define it, e.g. as "Deep Graph Network".

There were some spelling mistakes, but I suggest finding and fixing one instance of "regularation" in the text.

---

### Decision · Program_Chairs · 2019-12-19

**Decision:**

Accept (Poster)

**Comment:**

The work proposes a graph convolutional network based approach to multi-agent reinforcement learning. This approach is designed to be able to adaptively capture changing interactions between agents. Initial reviews highlighted several limitations but these were largely addressed by the authors. The resulting paper makes a valuable contribution by proposing a well-motivated approach, and by conducting extensive empirical validation and analysis that result in novel insights. I encourage the authors to take on board any remaining reviewer suggestions as they prepare the camera ready version of the paper.